# Graft-infiltrating host dendritic cells play a key role in organ transplant rejection

Quan Zhuang[1,2,*], Quan Liu[1,3,*], Sherrie J. Divito[1,†], Qiang Zeng[1], Karim M. Yatim[1], Andrew D. Hughes[1,4], Darling M. Rojas-Canales[1,†], A. Nakao[1], William J. Shufesky[1], Amanda L. Williams[1], Rishab Humar[1], Rosemary A. Hoffman[1,5], Warren D. Shlomchik[1,5,6], Martin H. Oberbarnscheidt[1,**], Fadi G. Lakkis[1,5,6,**] & Adrian E. Morelli[1,5,**]

Successful engraftment of organ transplants has traditionally relied on preventing the activation of recipient (host) T cells. Once T-cell activation has occurred, however, stalling the rejection process becomes increasingly difficult, leading to graft failure. Here we demonstrate that graft-infiltrating, recipient (host) dendritic cells (DCs) play a key role in driving the rejection of transplanted organs by activated (effector) T cells. We show that donor DCs that accompany heart or kidney grafts are rapidly replaced by recipient DCs. The DCs originate from non-classical monocytes and form stable, cognate interactions with effector T cells in the graft. Eliminating recipient DCs reduces the proliferation and survival of graft-infiltrating T cells and abrogates ongoing rejection or rejection mediated by transferred effector T cells. Therefore, host DCs that infiltrate transplanted organs sustain the alloimmune response after T-cell activation has already occurred. Targeting these cells provides a means for preventing or treating rejection.

[1] Thomas E. Starzl Transplantation Institute, Department of Surgery, University of Pittsburgh School of Medicine, Pittsburgh, Pennsylvania 15261, USA. [2] Center for Organ Transplantation, 3rd Xiangya Hospital, Central South University, Changsha 410083, China. [3] Department of Cardiovascular Surgery, The Second Affiliated Hospital of Harbin Medical University, Harbin, 150081, China. [4] Physician Scientist Training Program, University of Pittsburgh School of Medicine, Pittsburgh, Pennsylvania 15261, USA. [5] Department of Immunology, University of Pittsburgh School of Medicine, Pittsburgh, Pennsylvania 15261, USA. [6] Department of Medicine, University of Pittsburgh School of Medicine, Pittsburgh, Pennsylvania 15261, USA. * These authors contributed equally to this work. ** These authors jointly supervised this work. † Present Addresses: Department of Dermatology, Brigham & Women's Hospital, Harvard Medical School, Boston, Massachusetts 02115, USA (S.J.D.); Centre for Clinical and Experimental Transplantation, The Royal Adelaide Hospital, Adelaide, South Australia 5000, Australia (D.M.R.-C.). Correspondence and requests for materials should be addressed to M.H.O. (email: mho6@pitt.edu) or to F.G.L. (email: lakkisf@upmc.edu) or to A.E.M. (email: morelli@pitt.edu).

mprovement in organ allograft survival over the past 30 years can be attributed to the development of potent inhibitors of T-cell activation and proliferation. Despite these advances, a substantial proportion of transplanted organs are still rejected[1]. Rejection results from incomplete inhibition of recipient T cells that recognize donor alloantigens, leading to the generation of effector and memory T cells[2]. Since effector and memory T cells are more difficult to suppress or eliminate than naive T cells[3–6], rejection becomes increasingly difficult to treat or prevent once T-cell priming has occurred. This is borne out by clinical data showing that patients with pre-existing anti-donor memory T cells or those who experience acute rejection are at significantly increased risk of graft loss[7–9]. Therefore, understanding the factors that sustain the alloimmune response beyond initial T-cell activation is necessary for developing more effective anti-rejection therapies.

A key cell that participates in T-cell activation is the dendritic cell (DC). DCs activate T cells by presenting antigenic peptides in the context of MHC molecules to the T-cell receptor (TCR), and by providing co-stimulatory signals required for T-cell proliferation and differentiation[10]. In organ transplantation, donor DCs that accompany the graft migrate to the recipient's secondary lymphoid tissues[11–13]. There they initiate the alloimmune response by presumably engaging host alloreactive T cells or by transferring donor alloantigens to recipient (host) DCs[14–16]. In the latter case, alloantigens (for example, non-self MHC molecules) are transferred intact (semi-direct antigen presentation or cross-dressing) or are taken up and presented to recipient T cells as non-self peptides bound to self-MHC molecules (indirect antigen presentation or cross-priming)[17,18]. Although transplanted organs are eventually depleted of donor DCs, they are amply reconstituted with recipient DCs after transplantation[19–22]. What role the latter cell population plays is unclear. One possibility is that recipient DCs enhance alloimmunity by capturing donor antigens in the graft and activating additional T cells in secondary lymphoid tissues[22]. Another significant possibility is that they exert their function locally by engaging effector T cells within the graft.

In this study, we tested the hypothesis that recipient DCs play a key role in rejection by forming cognate interactions with effector T cells in the graft and sustaining T-cell responses beyond initial T-cell activation in secondary lymphoid tissues. We utilized flow cytometry, immunohistology and intravital microscopy to investigate donor DC replacement by host DCs in mouse heart and kidney grafts; to determine the phenotype, function and origin of the host DCs; and to study their interactions with effector T cells in the graft. We then performed in vivo DC depletion experiments to establish their role in allograft rejection.

## Results

### Replacement of donor DCs by host DCs in heart grafts

Donor-derived DCs exit organ allografts after transplantation and are replaced by recipient DCs. This observation is based on classical histological studies that are limited in their phenotypic and functional characterization of DCs[19–21]. We therefore analysed myeloid cell populations in mouse heart grafts by flow cytometry and conducted functional studies on isolated graft DCs. DCs were identified as $Lin^{neg}Ly6G^{neg}CD11c^{+}MHC-II^{+}$ leukocytes, and recipient and donor DCs were distinguished by CD45.1 and CD45.2 expression, respectively (Supplementary Fig. 1). We observed that recipient DCs populate both syngeneic and allogeneic grafts, while donor-derived DCs dissipate quickly after transplantation (Fig. 1a). Recipient DCs represented $>85\%$ of DCs in the grafts on day 1 and $>95\%$ by day 7 (Fig. 1a). Comparable

magnitude and kinetics of DC replacement were observed in fully allogeneic and semi-allogeneic (BALB/c × B6)F1 (F1) grafts, but recipient DC number was significantly less in syngeneic grafts or in allografts transplanted to mice treated with the T-cell immunosuppressant cyclosporine A (Fig. 1a). Increase in recipient DCs over time did not result from in situ proliferation because $CD11c^{+}$ cells in the allografts were $Ki67^{neg}$ (Fig. 1b). Thus, donor DCs in heart grafts are rapidly replaced by recipient DCs. Moreover, an ongoing anti-donor alloimmune response significantly increases recipient DC accumulation.

To validate the phenotype and function of recipient DCs, we compared $CD11c^{+}$ and $CD11c^{neg}$ myeloid cell populations in the graft. We found that the majority of $CD45.1^{+}Lin^{neg}Ly6G^{neg}$ recipient cells are $CD11b^{+}$ and contain both $CD11c^{+}$ and $CD11c^{neg}$ populations (Fig. 1c). Most $CD11c^{+}$ cells were $Ly6C^{lo}$ (Fig. 1c) and differed from the $CD11c^{neg}$ population by higher expression of MHC-II and CD86, weaker staining with the macrophage marker F4/80, and undetectable CCR2 expression (Supplementary Fig. 2a)—a phenotype consistent with that of a DC. Poor F4/80 staining of $CD11c^{+}$ cells was confirmed by immunohistology (Supplementary Fig. 2b). An alternative flow analysis strategy in which $CD45.1^{+}Lin^{neg}Ly6G^{neg}CD11c^{+}$ $MHC-II^{+}$ cells were gated first confirmed that $\sim 90\%$ were $F4/80^{lo}CD11b^{+}$, likely representing monocyte-derived DC (mono-DC), and $\sim 10\%$ were $F4/80^{lo}CD11b^{neg}$, likely representing conventional DC (cDC; Supplementary Fig. 2c). Unlike their $CD11c^{neg}$ counterparts, recipient-derived ($CD45.1^{+}$ $CD45.2^{neg}$) $CD11c^{+}$ cells sorted from day 7 grafts triggered proliferation of and IFN-$\gamma$ secretion by allogeneic naive CD4 and CD8 T cells in a direct mixed leukocyte culture (MLC; Fig. 1c) and, when cultured alone, produced significantly more IL-12p70 (Fig. 1d). Sorted recipient-derived ($CD45.1^{+}$ $CD45.2^{neg}$) $CD11c^{+}$ cells also induced proliferation of 1H3.1 and 2C T cells via indirect and semi-direct (cross-dressing) antigen presentation, respectively (Fig. 1e). 1H3.1 cells are $CD4^{+}$ TCR-transgenic (tg) cells specific for BALB/c $IE\alpha_{52-68}$ presented in context of B6 $IA^{b}$ (ref. 23), while 2C cells are $CD8^{+}$ TCR-tg cells that recognize intact BALB/c MHC-I molecule $H-2L^{d}$ (ref. 24). Further characterization of the $CD11c^{neg}$ population revealed that it consists of $Ly6C^{hi}F4/80^{hi}CCR2^{+}$ and $Ly6C^{lo}F4/80^{lo}CCR2^{neg}$ subsets (Fig. 1c; Supplementary Fig. 2a), the former secreting the highest level of TNF$\alpha$ (Supplementary Fig. 2d). Both $CD11c^{+}$ and $CD11c^{neg}$ cells were $CD103^{neg}$ and $CD8\alpha^{neg}$, but expressed CD115 and CCR7 (Supplementary Fig. 2a). Therefore, the vast majority of recipient $CD11c^{+}$ cells that accumulate in heart allografts are $CD11b^{+}CD8\alpha^{neg}CD103^{neg}$ DCs with potent allostimulatory and Th1-polarizing functions, whereas $CD11c^{neg}$ cells likely correspond to monocytes or macrophages with inflammatory functions, but poor allostimulatory capabilities.

### Host DCs in heart grafts stem from non-classical monocytes

The data so far suggest that the majority of recipient DCs in the graft derive from monocytes because they express CD11b and CD115. To establish their origin and test whether they stem from classical/inflammatory ($CCR2^{+}CX_{3}CR1^{lo}Ly6C^{hi}$) or non-classical/patrolling ($CCR2^{neg}CX_{3}CR1^{hi}Ly6C^{lo}$) monocytes, we first investigated heart allografts transplanted to wild-type (WT) and $CCR2^{KO}$ mice. In the latter, classical monocytes are retained in the bone marrow due to absent CCR2 signalling[25]. As shown by flow cytometry (Fig. 2a) and immunohistology (Fig. 2b), grafts transplanted to $CCR2^{KO}$ or WT mice were equally populated with recipient DCs despite significant reduction in intragraft and circulating classical monocytes in $CCR2^{KO}$ recipients, indicating that classical monocytes are not an important precursor of graft DCs. Next we performed adoptive transfer experiments in which

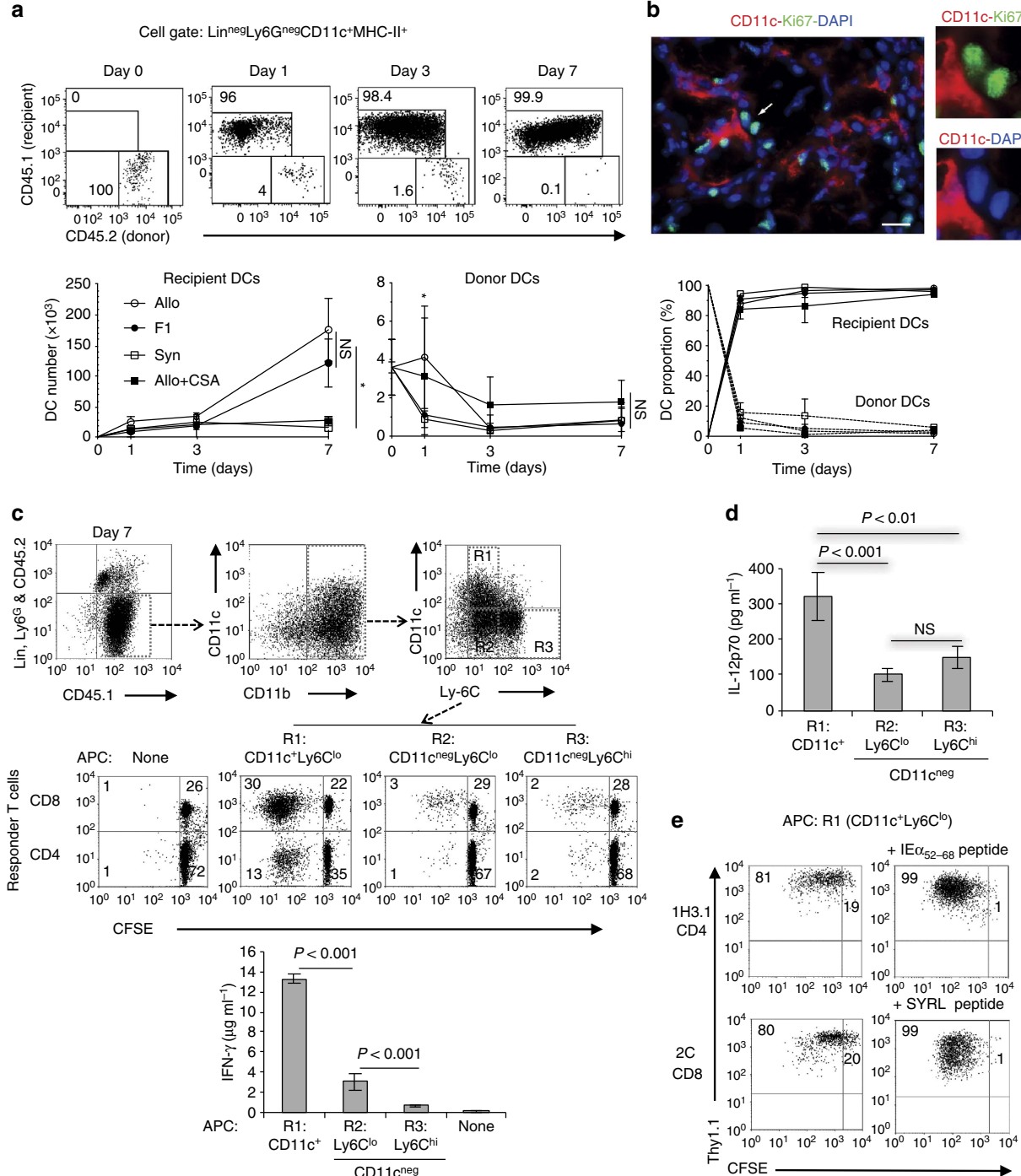

**Figure 1 | Replacement of donor DCs by host DCs in heart grafts. (a)** CD45.2 BALB/c (Allo), (B6 x BALB/c) F1 (F1) or B6 (Syn) heart grafts were transplanted to CD45.1 B6 mice. Intragraft recipient (CD45.1) and donor (CD45.2) DCs were analysed by flow cytometry at indicated time points (day 0, day 1, day 3 and day 7). Representative flow plots and line graphs are shown (mean ± s.d., 3 or 4 mice per time point). Full gating shown in Supplementary Fig. 1. CSA, Cyclosporine A. NS, not significant. *$P < 0.01$ (one-way analysis of variance (ANOVA)). In the case of donor DC enumeration significant change was detected in DC number over time but not between groups. **(b)** Representative photomicrograph of a tissue section of one of 3 BALB/c heart allografts harvested on day 7 and stained for CD11c (red) and proliferation marker Ki67 (green). Cell nuclei are stained with DAPI (blue). The CD11c+ DCs (in red) were Ki67neg. The arrow indicates Ki67 expression, as an endogenous control of Ki67 labelling, in the nuclei of CD11cneg cells shown in detail in the insets. Magnification × 200. Scale bar, 15 μm. **(c)** Characterization of LinnegLy6Gneg, recipient (CD45.1+) myeloid cells in BALB/c heart allografts on day 7 after transplantation. Cells were analysed by flow cytometry (top panels) and CD11c+ and CD11cneg subsets were sorted and tested for their ability to induce proliferation (centre panels) and IFN-γ production (bar graph) in naive, allogeneic T cells in the direct MLC. Proliferation was measured by CFSE dilution and IFN-γ by ELISA of culture supernatants on day 5 of MLC. One representative experiment out of two is shown. $P$ values were generated by one-way analysis of variance (ANOVA). Bar graphs are mean ± s.d. (3 mice per group). **(d)** IL-12p70 production by sorted CD11c+ and CD11cneg subsets cultured for 24 h. Cells sorted from 2 different BALB/c heart allografts on day 7 were pooled and ELISA was performed in duplicates on the culture supernatant. The experiment was performed twice (4 mice total). Bar graphs are mean ± s.d. of 4 measurements (2 experimental × 2 technical replicates). $P$ values were generated by one-way ANOVA. **(e)** Ability of sorted CD11c+ subset to stimulate proliferation of CD4+ 1H3.1 and CD8+ 2C T cells *ex vivo* in the absence (left panels) or presence (right panels) of cognate peptide added to the culture (3 mice per group).

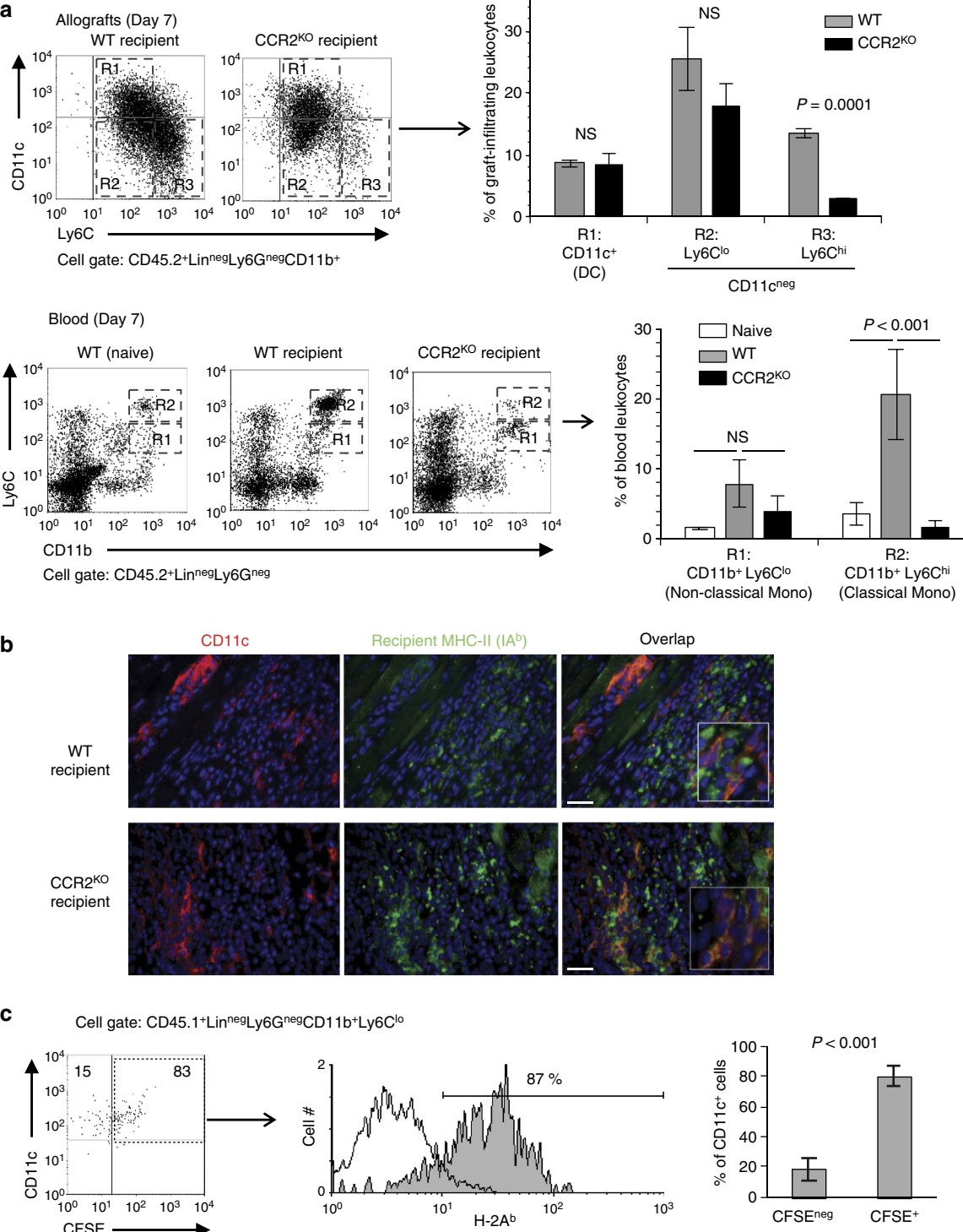

**Figure 2 | Origin of recipient DCs in heart allografts.** (**a**) BALB/c heart allografts were transplanted to B6 WT or CCR2$^{KO}$ mice. Grafts and recipient blood were collected on day 7. DC (CD11c$^+$) and non-DC (CD11c$^{neg}$) subsets were identified and enumerated in the graft (>95% at this time point are recipient-derived as shown in Fig. 1a), and recipient classical (Ly6C$^{hi}$) and non-classical (Ly6C$^{lo}$) monocyte subsets were enumerated in the blood. As control, monocyte subsets in blood of untransplanted (naive) B6 mice were also analysed. Bars represent mean ± s.d. (N = 3 or 4 mice per group). P values were generated by one-way analysis of variance (ANOVA). (**b**) Representative photomicrographs out of three allografts per group depicting graft tissue stained for CD11c (red) and recipient MHC-II (IA$^b$) (green). Overlay (orange) demonstrates recipient origin of DCs in grafts removed from either WT or CCR2$^{KO}$ mice. Magnification ×200. Scale bar, 50 μm. (**c**) Equal numbers of unlabelled, classical (Ly6C$^{hi}$) and CFSE-labelled, non-classical (Ly6C$^{lo}$) CD45.1$^+$ B6 monocytes were co-transferred to CD45.2$^+$ B6 mice 5 days after receiving CD45.2$^+$ BALB/c heart allografts. Flow analysis of graft-infiltrating leukocytes was performed 2 days later. Flow plots show the differentiation of transferred (CD45.1$^+$) monocytes into DCs (CD11c$^+$MHC-II$^+$) in the graft, the majority of which were CFSE$^+$ (derived from non-classical monocytes). Bar graph is mean ± s.d. (N = 3 mice). P values were calculated with two-sided Student's t-test.

equal numbers of unlabelled, classical and CFSE-labelled, non-classical CD45.1$^+$ B6 monocytes were co-transferred to CD45.2$^+$ B6 recipients of CD45.2$^+$ allografts. Flow analysis of graft-infiltrating leukocytes 2 days after monocyte transfer revealed that the majority of CD45.1$^+$ DCs were CFSE$^+$ (Fig. 2c), implying that recipient DCs in the graft stem predominantly from non-classical monocytes.

**Replacement of donor DCs by host DCs in kidney grafts**. We next examined whether replacement of donor DCs by recipient mono-DCs also occurs in kidney grafts, which harbour a more extensive network of DCs than heart grafts[26]. We transplanted kidney allografts from CD45.2 F1 CD11c-YFP (yellow fluorescent protein) mice to CD45.1 B6 CX3CR1$^{gfp/+}$ recipients in which one locus of the fractalkine receptor has been replaced with the GFP gene[27]. In these recipients, mono-DCs express high levels of green fluorescent protein (GFP). Grafts were imaged by multiphoton intravital microscopy and collected for flow analysis at end of imaging. As shown by microscopy (Fig. 3a), and confirmed by flow analysis (Fig. 3b), donor DCs (YFP$^+$) were replaced by a greater number of recipient mono-DCs (GFP$^+$) by day 7 after transplantation. The monocyte origin of recipient DCs was established by imaging kidney allografts transplanted to CD11c-YFP x CX3CR1$^{gfp/+}$ mice in which mono-DCs are double-colored (YFP$^+$GFP$^+$) while cDCs are single-colored (YFP$^+$). As shown in Fig. 3c, majority of DCs present on day 7 were double-coloured (depicted in red), strongly suggesting that they belong to the monocyte lineage. Single-coloured YFP$^+$ cells were a minority while single-coloured GFP$^+$ cells were rare and did not have dendritic cell morphology. Flow cytometry confirmed conspicuous predominance of mono-DCs over cDCs in kidney allografts (Fig. 3c).

**Host DCs in the graft form cognate contacts with T cells**. The large number of recipient DCs in the graft and their potent allostimulatory capacity suggest that they could participate in the local effector immune response. This hypothesis is supported by our observation that DCs capture antigen-specific effector T cells inside graft capillaries and mediate their trans-endothelial migration[28]. To test whether recipient DCs continue to form cognate interactions with effector T cells in the graft's interstitium, we transplanted F1 Act-OVA kidneys to chimeric B6 mice that harbour equal numbers of H2-Kb$^{-/-}$ (YFP$^+$) and H2-Kb$^{+/+}$ (GFP$^+$) DCs. OT-I effector T cells were transferred 1 week after transplantation and multiphoton intravital microscopy of the transplanted kidney performed 1 day later. Since the OVA peptide recognized by OT-I cells is presented by H-2K$^b$, cognate T-cell–DC interactions could be discerned by quantifying OT-I contacts with H2-K$^{b+/+}$ and H2-K$^{b-/-}$ DCs in the same graft. OT-I effectors were readily visible in the renal cortex one day after transfer (Fig. 4a, Supplementary Movie 1), >90% had transmigrated into the interstitium (Fig. 4b), and >80% were contacting (>2 min duration) H2-K$^{b+/+}$ or H2-K$^{b-/-}$ DCs (Fig. 4c, Supplementary Movie 2). However, a greater proportion of OT-I effectors contacted H2-K$^{b+/+}$ than H2-K$^{b-/-}$ DCs (Fig. 4d), and the mean contact time with H2-Kb$^{+/+}$ DCs was significantly longer (18 versus 6 min, respectively; Fig. 4e). The increased contact time correlated with lower mean velocity and higher arrest coefficient for OT-I effectors contacting H-2Kb$^{+/+}$ DCs (Fig. 4e). These results indicate that cognate antigen recognition stabilizes effector T-cell–DC interactions. To directly test the role of recipient DCs in arresting effector T cells in the graft, we transplanted F1 Act-OVA kidneys to B6 CD11c-DTR→B6 bone marrow chimeras and depleted DCs by injecting diphtheria toxin (DT) after OT-I transfer. As shown in Fig. 4f, DC depletion significantly increased the velocity and decreased the arrest coefficient of effector T cells in the graft. Therefore, recipient DCs engage infiltrating effector T cells in a cognate fashion, leading to their stable arrest.

**Host DCs promote effector T-cell-mediated rejection**. To test the functional significance of effector T-cell–DC interactions in the graft, we investigated the effect of recipient DC depletion on heart allograft survival in two experimental models. In the first model, we initiated DC depletion 5 days after transplantation, at which time effector T cells generated in secondary lymphoid tissues had already infiltrated the graft. In the second model, we performed DC depletion in splenectomized LTβR$^{KO}$ recipients, which lack all secondary lymphoid tissues and cannot mount a primary alloimmune response[15,29], and studied graft rejection after transferring effector/memory (CD44$^{hi}$) T cells primed against donor antigens.

As shown in Fig. 5a, delayed depletion of DCs in B6 CD11c-DTR→B6 bone marrow chimeras by repeated DT administration starting on day 5 after transplantation and continued every other day until graft rejection, significantly prolonged allograft survival compared with control B6→B6 chimeras that received the same DT regimen or to control B6 CD11c-DTR→B6 bone marrow chimeras injected with PBS instead of DT (MST = 30 versus 10 days in both cases). Analysis of the grafts on day 7 confirmed DC depletion in the CD11c-DTR group (Fig. 5b), and showed decreased number, decreased proliferation, and increased apoptosis of infiltrating CD4 and CD8 T cells (Fig. 5c). Reduced in situ T-cell proliferation after DC depletion was confirmed by immunostaining graft tissue for Ki67 and the mitotic marker Histone 3 pS10 (Fig. 5d). In contrast to these results, allografts transplanted to B6 CCR2$^{KO}$ mice, in which CD11b$^+$CD11c$^{neg}$ cells, but not CD11b$^+$CD11c$^+$ cells (which represent mono-DCs) are reduced in the graft (Fig. 2a), were promptly rejected with normal proliferation of infiltrating T cells (Supplementary Fig. 3). These findings imply that recipient DCs play an important role in rejection by promoting the proliferation and survival of effector T cells within the graft.

Next we transplanted BALB/c hearts to splenectomized B6 CD11c-DTR→B6 LTβR$^{KO}$ bone marrow chimeras and transferred effector/memory CD4$^+$ and CD8+ T cells from BALB/c-sensitized B6 mice 2 days later. Recipients were left untreated or were injected with DT beginning on day of transplantation or 10 days later, and every other day thereafter until graft rejection. Control groups included splenectomized CD11c-DTR→B6 LTβR$^{KO}$ chimeras that did not receive effector T cells (one group treated with DT and one not), and splenectomized WT→B6 LTβR$^{KO}$ chimeras that received T cells and were treated with DT starting on day of transplantation and every other day until graft rejection. As shown in Fig. 6a, mice that did not receive effector T cells did not reject their grafts, affirming the inability of these mice to mount a primary alloimmune response. In contrast, transferred effector/memory T cells precipitated allograft rejection in DC-replete CD11c-DTR chimeras (no DT administered; MST = 29 days) and in DC-replete WT chimeras treated with DT (MST = 31 days), but not in DC-depleted CD11c-DTR chimeras (MST > 75 days; $P < 0.0001$; Fig. 6a). Delaying DT administration until day 10 after transplantation prolonged allograft survival in CD11c-DTR chimeras by ∼14 days (MST = 44 days), but did not reach statistical significance. Immunohistology demonstrated that grafts collected from the DT-treated group lacked CD11c$^+$ cells and had a significant reduction in CD4$^+$ and CD8$^+$ T-cell infiltrates (Fig. 6b).

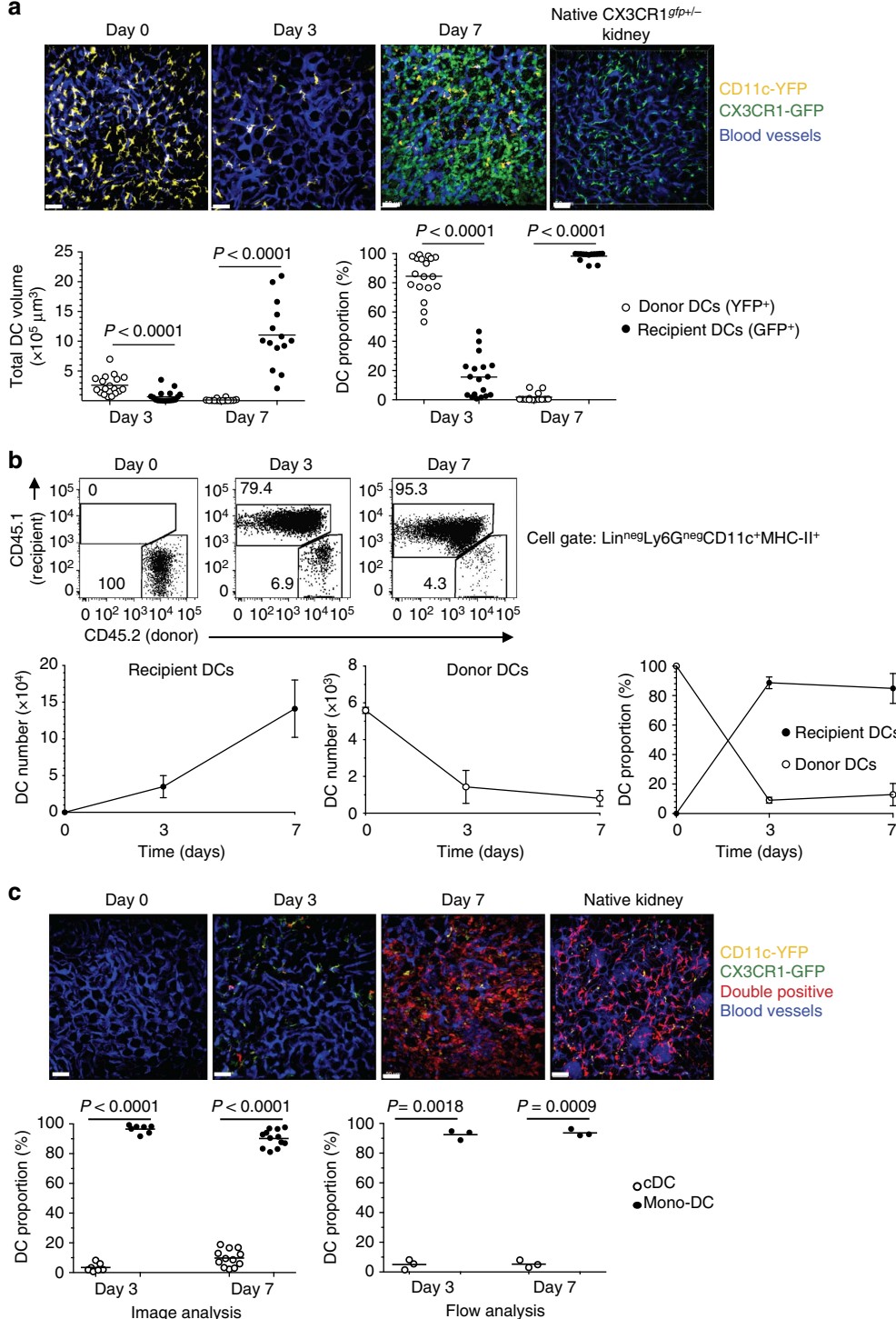

**Figure 3 | Rapid replacement of donor DCs in kidney allografts by recipient monocyte-derived DCs.** (**a**) F1 CD11c-YFP kidney allografts (CD45.2) were transplanted to B6 CX3CR1$^{gfp/+}$ recipients (CD45.1) and two-photon intravital imaging was performed at indicated time points. Representative volume-rendered images are shown in the top panels and quantitation of DC volume and DC proportions in the graphs below. Panel on far left (Day 0) is an image of a native CD11c-YFP kidney before transplantation. Scale bar, 50 μm. Frame indicates dimensions of the 3D volume. $N = 14$–18 images from 3 or 4 mice per time point. (**b**) Cells were extracted from allografts at the end of imaging and analysed by flow cytometry. Representative flow plots and line graphs (mean ± s.d., $N = 3$ mice per time point) of absolute number and proportion of recipient and donor DCs are shown. (**c**) F1 kidney allografts not transgenic for a fluorescent protein were transplanted to B6 CX3CR1$^{gfp/+}$xCD11c-YFP recipients and two-photon intravital imaging was performed at the indicated time points. Representative volume-rendered images are shown in the top panels. Panel on far left (Day 0) is an image of a native non-transgenic F1 kidney before transplantation. Graphs show proportions of DC subsets by volumetric image analysis (left) and flow analysis (right). In the image analysis, single positive (YFP$^+$) cells represented cDCs, and double-positive (YFP$^+$GFP$^+$) cells mono-DCs. In the flow analysis, cDCs (CD11b$^{neg}$) and mono-DCs (CD11b$^+$) were distinguished based on CD11b expression on CD45.1$^+$Lin$^{neg}$Ly-6G$^{neg}$CD11c$^+$MHC-II$^+$ cells. Scale bar, 50 μm. $N = 7$–12 images from 3 mice per time point. In **a** and **c**, horizontal lines are mean values and $P$ values were calculated using two-sided Student's *t*-test.

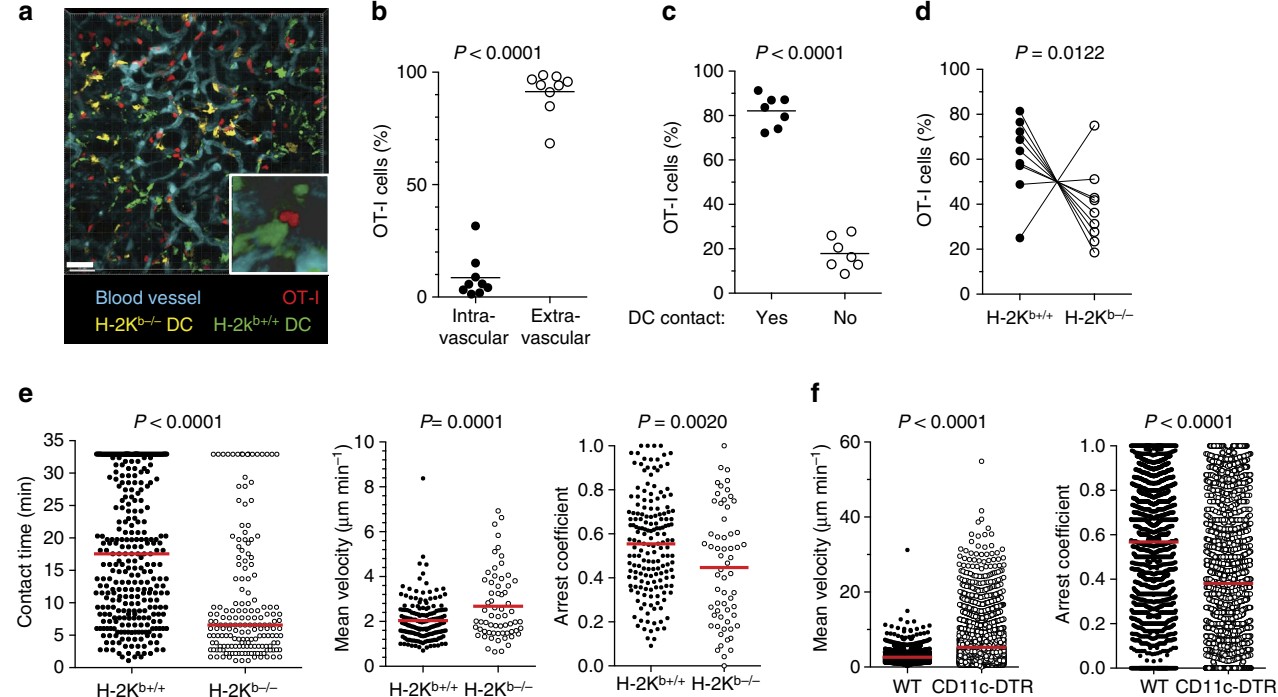

**Figure 4 | Interactions between recipient DCs and effector T cells in kidney allografts.** F1.Act-OVA kidneys were transplanted to B6 mixed bone marrow chimeras that harbour equal numbers of H2-K$^{b-/-}$ (YFP$^+$) and H2-K$^{b+/+}$ (GFP$^+$) DCs. OT-I effector T cells were transferred 6 days after transplantation and intravital two-photon microscopy was performed 1 day later. (**a**) Representative, volume-rendered image of kidney allograft with inset showing an OT-I cell (red) in close contact with a DC (green). Scale bar, 50 μm. (**b**) Number of OT-I cells located inside (intravascular) or outside (extravascular) of blood vessels. (**c**) Proportion of OT-I cells making >2 min contact with either type of DC in the graft. (**d**) Proportion of OT-I cells in contact with H-2K$^{b+/+}$ versus H-2K$^{b-/-}$ DCs. Lines join data points from same image volumes. (**e**) Contact time, mean velocity, and arrest coefficient of OT-I cells interacting with H-2K$^{b+/+}$ versus H-2K$^{b-/-}$ DCs during entire time-lapse recording (~33 min). N = 11 image data sets (movies) from five mice. (**f**) F1.Act-OVA kidney grafts were transplanted to CD11c-DTR or WT bone marrow chimeric B6 mice. DT was administered to both groups of mice on days 7 and 9, OT-I T cells were adoptively transferred on day 7, and imaging performed on day 10 after transplantation. N = 9 image data sets (movies) from 4 mice/group. In **b**–**f**, horizontal lines are mean values and P values were calculated using two-sided Student's t-test.

To ensure that differences in graft outcomes between DT-treated and untreated mice were not due to a difference in engraftment of transferred T cells, we tested the effect of repeated DT administration on the number and function of effector T cells transferred to splenectomized CD11c-DTR→LTβR$^{KO}$ chimeras. As shown in Supplementary Fig. 4, 21 days after transfer CD44$^{hi}$ T cells were readily detected in the bone marrow and blood despite continuous DC depletion, and neither their number nor capacity to produce IFN-γ were reduced compared with untreated mice. Altogether, the findings indicate that recipient DCs that infiltrate the graft play a key role in rejection.

## Discussion

The results reported in this study provide novel insights into the nature, origin and function of host DCs that infiltrate organ transplants. First, we established that the majority of recipient DCs that replace donor DCs in heart and kidney grafts are non-conventional CD11b$^+$CD11c$^+$ DCs that produce IL-12, originate from non-classical monocytes, and are phenotypically and functionally distinct from other monocyte-derived cells in the graft. These findings are consistent with reports showing that non-classical monocytes are the principal source of CD11c$^+$ cells at sites of sterile inflammation (for example, in atherosclerotic plaques)[30,31], but differ from studies in infection models where the majority of DCs have an inflammatory, TNFα-producing phenotype and arise from classical monocytes[32–34]. These differences are likely due to distinct stimuli that drive monocyte

differentiation in separate inflammatory settings. It is also possible that transplantation presents a unique situation in which allogeneic non-self, working in concert with or in addition to inflammation, triggers monocyte differentiation to DCs. We have recently shown that IL-12 producing DCs with monocyte lineage markers accumulate and persist over time in allogeneic but not syngeneic grafts transplanted to mice that lack adaptive immune cells[35].

Second, we demonstrated that graft-infiltrating, host DCs preferentially induce T-cell differentiation to a Th1 phenotype, which is the dominant T-cell phenotype in rejection. Antigen presentation by host DCs was not restricted to cross-priming indirectly alloreactive T cells, but was also effective at priming directly alloreactive T cells. Activation of the latter by host DCs is possible if DCs were 'cross-dressed' with intact donor MHC molecules in the graft. Although not formally shown here, cross-dressing has been described in vitro and in vivo and likely participates in immune responses to viruses, bone marrow transplants and organ transplants[17,36–39]. Therefore, host DCs that replace donor DCs in the graft are not limited to engaging indirectly alloreactive T cells, which represent a minute fraction of the host's alloreactive T-cell repertoire, but can also present non-self MHC molecules to the much larger population of directly alloreactive T cells.

Third, we provided direct visual evidence that host DCs engage effector T cells in the graft. Most effector T cells that had transmigrated into the graft made stable contacts with DCs. The contacts were more numerous and stable with DCs that

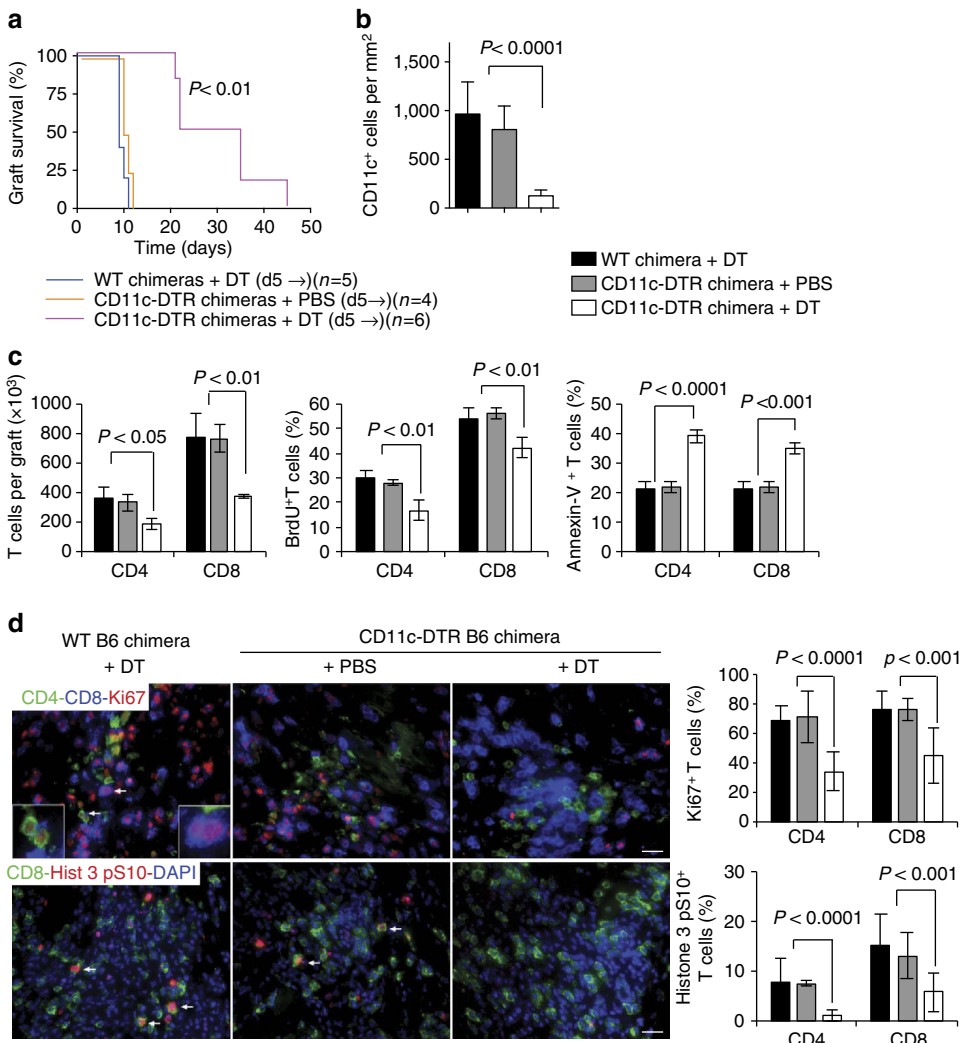

**Figure 5 | Effect of delayed recipient DC depletion on allograft survival and T-cell proliferation and apoptosis in the graft.** (**a**) BALB/c heart allografts were transplanted to chimeric B6 mice that received either B6 WT or CD11c-DTR bone marrow. Mice were injected with DT or PBS every other day starting on day 5 after transplantation until day of rejection (d 5→). Allograft survival was monitored by daily palpation of heart contractions. Number of mice per group is between parentheses. Kaplan–Meier graft survival plot is shown. P value was calculated by the log-rank method. (**b–d**) Transplantation and DT treatment were performed as in **a** except that all grafts were sacrificed on day 7 after transplantation (2 days after a single DT injection) for immunostaining and flow cytometry. Quantitation of intragraft DCs by immunostaining of CD11c is shown in **b**. The number of intragraft CD4 and CD8 T cells and proportion undergoing proliferation (BrdU$^+$) or apoptosis (Annexin-V$^+$) is shown in **c**. T cells undergoing proliferation *in situ* were detected and quantified by staining for Ki67 or Histone 3 pS10 (**d**). White arrows point to T cells that are positive for the proliferation markers. Images are representative out of 5–6 fields, at ×200 magnification, analysed on tissue sections of each of four allografts per group. Magnification ×200. Scale bar, 20 μm. Bar graphs are mean ± s.d. N = 4 mice per group. P values were generated by one-way analysis of variance (ANOVA).

presented cognate antigen than those that did not. Our observation that OT-I effectors continue to make interactions with H-2Kb$^{-/-}$ DCs (Fig. 4), some of which are very stable, could suggest that cross-dressing of these DCs with H-2Kb molecules carrying the OVA peptide has taken place. Alternatively, these interactions, especially the shorter duration ones, could be mediated by chemokines. Altogether, these observations extend our previous findings that DCs capture anti-donor effector T cells within capillaries and promote their trans-endothelial migration[28]. Therefore, graft-infiltrating, host DCs continue to engage effector T cells even after the latter have transmigrated into the graft parenchyma.

Finally, we demonstrated the functional significance of interactions between graft DCs and effector T cells by depleting DCs in heart transplant recipients. We showed that host DC depletion reduces arrest, proliferation, and survival of effector

T cells *in situ*. In graft survival experiments, host DC depletion interrupted ongoing rejection in WT recipients and prevented rejection exclusively mediated by effector T cells in mice that lacked secondary lymphoid tissues. These data strongly point to the graft as the site of productive host DC-effector T-cell interactions that drive rejection.

An additional mechanism by which graft-infiltrating, host DCs could promote rejection is through migration to secondary lymphoid tissues where they prime more T cells. Celli *et al.*[22] showed that host DCs that infiltrate skin allografts home to draining lymph nodes and cross-prime CD8$^+$ T cells on re-transplanting the graft to a second recipient. Since in this scenario retransplantation was a powerful trigger for DC exit from the graft, it remains to be determined whether graft-infiltrating, recipient DCs migrate to secondary lymphoid tissues from an organ that had been transplanted only once. It is also possible that recipient

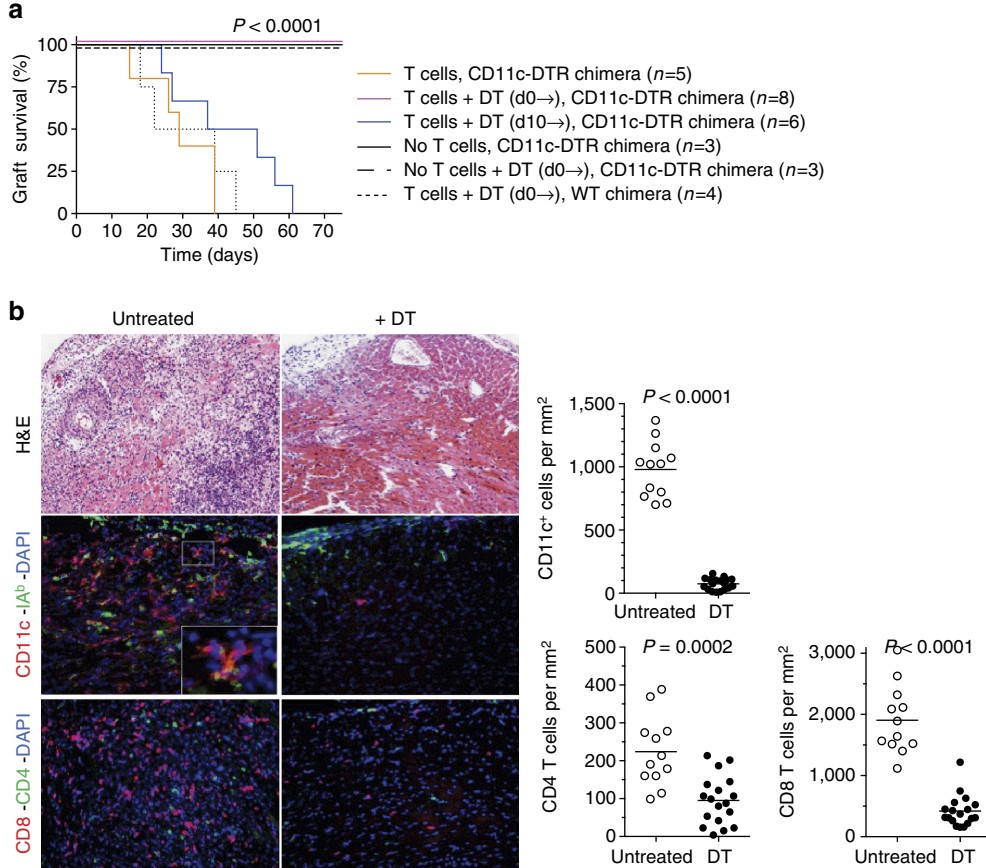

**Figure 6 | Effect of recipient DC depletion on rejection mediated by effector T cells.** (**a**) BALB/c heart allografts were transplanted to splenectomized, CD11c-DTR→B6 LTβR[KO] bone marrow chimeras that lack all secondary lymphoid tissues. Experimental groups received effector T cells on day 2 after transplantation and were either left untreated or treated with DT starting at time of transplantation (d0→) or 10 days after transplantation (d10→). DT was injected every other day until the day of graft rejection. Control groups included splenectomized CD11c-DTR→B6 LTβR[KO] chimeras that did not receive effector T cells (one group treated with DT and one not), and splenectomized WT→B6 LTβR[KO] chimeras that received T cells and were treated with DT starting on day of transplantation. Allograft survival was monitored by daily palpation of heart contractions. Kaplan–Meier graft survival plot is shown. P value was determined by log-rank test. (**b**) Infiltration of heart allografts with recipient DC (CD11c[+]IA[b+] cells), CD4, and CD8 T cells in untreated and DT-treated (+DT) mice that received effector T cells on day 2. Representative photomicrographs out of three fields, at ×200 magnification, analysed on tissue sections of each of 4–6 allografts per group. Inset in photomicrograph is a DC stained positive for both CD11c and IA[b]. Horizontal lines in scatter plots represent mean for each group. Magnification ×100. Scale bar, 50 μm. P values were calculated using two-sided Student's t-test.

DCs recall alloreactive memory T cells in the graft. Memory T cells comprise a substantial fraction of the alloreactive T-cell repertoire[40], utilize the same mechanisms as effector T cells to access the graft[28], and cause rejection without the need for prior activation in secondary lymphoid tissues[41]. In infection models, DCs at non-lymphoid sites recall memory T cells[42], raising the likelihood that comparable events occur in organ transplants. Therefore, host DCs that populate transplanted organs could sustain the alloimmune response via multiple pathways.

The role of DCs in priming the immune response to transplanted organs is well-recognized[14,43]. This initial step occurs when naive T cells encounter alloantigen-bearing DCs in secondary lymphoid tissues[15]. Here we extended this fundamental paradigm by showing that DCs are also key to sustaining the alloimmune response outside secondary lymphoid tissues. This second step occurs when effector T cells encounter host DCs in the graft itself. The clinical implication is that rejection in patients who are already primed against donor alloantigens can be interrupted or prevented by targeting host DCs or their precursors, the monocytes. These patients are high-risk transplant recipients whose grafts fare poorly despite current immunosuppression.

## Methods

**Mice.** C57BL/6 (B6), BALB/cJ, B6.SJL-Ptprc[a]Pepc[b]/BoyJ (CD45.1), B6.129S4-CCR2[tm1Ifc]/J (CCR2[KO]), C57BL/6J-Tg(CAG-OVA)916Jen/J (CD45.2) (B6-OVA), and C57BL/6-Tg(TcraTcrb)1100Mjb/J (CD45.2) (OT-I) mice, B6.129S7-Rag1[tm1Mom]/J (B6 RAG[ko]), B6.129P(Cg)-Ptprc[a] Cx3cr1[tm1Litt]/LittJ (B6 CX₃CR1[gfp/gfp] CD45.1), B6.Cg-Tg(Itgax-Venus)1Mnz/J (CD11c-YFP) and B6.FVB-Tg (Itagx-DTR/EGFP)₅₇Lan/J (CD11c-DTR) were purchased from The Jackson Laboratory. B6 H2K[bko] mice were purchased from Taconic. OT-I mice were bred onto the Rag[ko] background. 1H3.1 and 2C TCR-tg mice on the Rag[ko] background were bred and maintained at the University of Pittsburgh animal facility. F1 mice were generated by cross-breeding C57/BL6J and BALB/cJ mice. F1-OVA and F1-CD11c-YFP mice were generated by cross-breeding BALB/cJ and C57BL/6J-Tg(CAG-OVA)916Jen/J mice and BALB/cJ and B6.Cg-Tg(Itgax-Venus)1Mnz/J mice, respectively. B6 H2k[bko] CD11c-YFP mice were generated by cross-breeding B6 H2k[bko] and B6 CD11c-YFP mice. CD11c-YFP x CX₃CR1[gfp/+] mice were bred by crossing B6 CD11c-YFP and homozygous CX₃CR1[gfp/gfp] mice. B6 LTβR[KO] mice were obtained from Yang-Xin Fu (University of Chicago) and maintained at the University of Pittsburgh animal facility. Both male and female mice were used except that we avoided male donor to female recipient combination because of H-Y antigen disparity. All mouse work was performed in compliance with ethical regulations and was approved by the Institutional Animal Care and Use Committee of the University of Pittsburgh.

**Reagents.** Monoclonal antibodies (Abs) were purchased from BD-PharMingen, eBioscience, Invitrogen, Jackson ImmunoResearch, Thermo Scientific, Millipore or Cedarlane. CFSE and 4-amino-5methylamino-2′7′-difluorofluorescenin diacetate

(DAF-FM diacetate) were from Molecular Probes-Invitrogen. Type IV collagenase and DT were from Sigma. Cyclosporine A (Sandimmune injection, Novartis) was administered daily at 40 mg kg$^{-1}$ intraperitoneally (i.p.).

**Surgical procedures and bone marrow chimeras.** Splenectomies and heterotopic transplantation of vascularized heart and kidney grafts were performed as follows[15,44,45]: Splenectomy—a 1–1.5 cm left flank incision midway between the last rib and the hip bone was made, followed by a peritoneal incision. Splenic vein and artery were tied off at the hilum and the spleen was removed. Animals recovered for at least 7 days before further experimental use. Heart transplantation—after a midline incision and sternotomy, the donor heart was perfused with 2 ml of 4 °C heparinized (100 U ml$^{-1}$) PBS via the inferior vena cava. The left superior vena cava and pulmonary veins were ligated near the atrium and divided distally to the ligatures. The ascending aorta and a segment of the innominate artery were mobilized. The aorta was cut close to the innominate artery before the bifurcation. The pulmonary artery was mobilized and transected at the point of bifurcation. The heart was collected and stored in 4 °C normal saline in sterile conditions. After a midline incision, the recipient's abdominal aorta and inferior vena cava were mobilized for a short segment from bifurcations of renal vessels to bifurcations of common iliac vessels. Lumbar veins behind the inferior vena cava were cauterized. The aorta and the pulmonary artery of the donor heart were anastomosed end-to-side to the recipient's abdominal aorta and vena cava. After reperfusion of the donor heart, the gut was replaced and allowed to resume its normal position around the grafted heart. Kidney transplantation—the donor kidney, renal artery, renal vein and ~1 cm segment of the abdominal aorta and inferior vena cava were recovered en bloc, perfused with ice-cold PBS containing 1% heparin and immersed in University of Wisconsin solution (Duramed Pharmaceuticals, Cincinnati, OH, USA) at 4 °C for 15 min until transplantation. The recipient was anaesthetized with 10 mg kg$^{-1}$ Xylazine and 100 mg kg$^{-1}$ Ketamine (Henry Schein, Melville, NY, USA) and end-to-side anastomosis of the aorta and vena cava of the donor to those of the recipient, respectively, was performed. Warm ischaemia time averaged 25 min. Heart graft rejection was defined as cessation of palpable heartbeat and was confirmed by histological analysis. Bone marrow chimeras were generated by irradiating recipient mice with 1,100 cGy in two doses 6 h apart from a Nordion Gamma Cell 40 caesium source. After irradiation the mice were injected intravenously (i.v.) with $1 \times 10^7$ donor bone marrow cells and given Sulfatrim in the drinking water for 1 week. After an 8-week reconstitution period, blood was phenotyped to verify appropriate reconstitution. Chimerism was consistently <90–95% in the blood.

**Isolation of graft-infiltrating leukocytes.** Mice were anaesthetized and perfused with 20–40 ml PBS + 0.5% heparin via the left ventricle until the fluid exiting the right ventricle did not contain any visible blood. Transplanted organs were then removed, cut into small fragments or homogenized using a GentleMACS tissue processor (Miltenyi), and incubated with 400 U ml$^{-1}$ type IV collagenase for 1 h at 37 °C, pipetting the fragments every 15 min. Single-cell suspensions were then obtained by passing the fragments through a 70 µm cell strainer, depleted of erythrocytes with NH$_4$Cl lysis buffer, rinsed in cold 0.010 M EDTA PBS and centrifuged on top of a Lympholyte-M (Cedarlane) density gradient (1,500g, 20 min, room temperature). Total recovered cells were counted using a hemocytometer or an automated cell counter (Beckman Coulter).

**Flow cytometric analysis.** Isolated cells were incubated with CD16/CD32 (2.4G2, 1:100) Ab to block FcR, and then labelled (30 min, 4 °C) with different combinations of fluorochrome-conjugated Abs specific for MHC class-I (AF6-88.5, 1:200) or II (AF6-120.1, 1:200) (H2$^b$), CD3 (145-2C11, 1:100), CD4 (RM4-5, 1:100), CD8α (53-6.7, 1:100), CD11c (HL3, 1:100), CD11b (M1/70, 1:200), CD19 (1D3, 1:100), CD45.1 (A20, 1:100), CD45.2 (104, 1:100), CD86 (GL1, 1:50), CD103 (M290, 1:100), CD115 (AFS98, 1:100), CCR2 (SA203G11, 1:100), CCR7 (4B12, 1:100), Ly6C (HK1.4, 1:800), Ly6G (1A8, 1:100), F4/80 (BM8, 1:200) or NK1.1 (PK136, 1:100), for flow analysis or cell sorting. Appropriate fluorochrome-conjugated isotype-matched Abs were used as negative controls. After staining, cells were fixed in 4% paraformaldehyde, acquired on a LSRII or LSR Fortessa flow cytometer (BD Biosciences) and analysed using FACSDiva (BD Biosciences), WinMDI 2.9 or FlowJo V9 software.

Annexin-V binding and BrdU incorporation were performed according to manufacturer's instructions (BD Biosciences). BrdU was injected i.p. 12 h before euthanasia. For detection of intracellular IFN-γ, cells were cultured for 5 h at 250,000 cells per well, in 96-well round-bottom plates with brefeldin A (10 µg ml$^{-1}$), PMA (20 ng ml$^{-1}$) and ionomycin (100 µM). After culture, cells were surface labelled with fluorochrome-conjugated CD4 and CD8 Abs, permeabilized with Cytofix/Cytoperm solution (eBiosciences) and labelled with PE-conjugated IFN-γ Ab (1:100).

**OT-I effector T cells.** CD8 OT-I T$_{eff}$ were generated in vivo similar to method previously described[46]. Briefly, B6 (CD45.1) mice were injected with 25 µg anti-DEC205-OVA hybrid antibody i.v. and 50 µg anti-CD40 Ab (FGK4.5) i.p. after adoptive transfer of $10^5$ OT-I Rag$^{KO}$ (CD45.2) splenocytes. Six days later,

OT-I T$_{eff}$ (CD45.2$^+$ CD8$^+$) were high-speed sorted using a BD FACS ARIA+ (BD Biosciences) cell sorter using the following exclusion channel: CD45.1$^+$ CD4$^+$ CD11b$^+$ CD11c$^+$ F4/80$^+$ CD45R/B220$^+$ (RA3-6B2, 1:200) Ly-76$^+$ (Ter-119, 1:200) CD16/32$^+$ CD49b$^+$ (DX5, 1:200). Purity was >95%. OT-I T$_{eff}$ were labelled with 2 µM Cell Tracker Red (20 min, 37 °C). $10^7$ OT-I T cells were transferred i.v. into kidney graft recipients before imaging.

**Immunostaining and microscopic analysis of graft tissue.** Graft tissue was snap-frozen in liquid nitrogen and stored at −80 °C until use. OCT embedded frozen tissues were sectioned by cryostat (10 µm). Tissue sections were fixed in 95% ethanol, and treated with 5% normal goat serum followed by the avidin/biotin blocking kit (Vector). Grafts were stained with (i) CD11c Ab (hamster, 1:100) and Ki67 (SP6) Ab (rabbit, 1:100) plus Cy3-anti-hamster IgG (1:400) and Cy2 anti-rabbit IgG (1:400); (ii) CD11c Ab (hamster, 1:100) and biotin-IA$^b$ (1:100), followed by Cy3-anti-hamster IgG (1:400) plus Cy2-streptavidin (1:400); (iii) CD11c Ab (hamster, 1:100) and biotin-CD45.1 (1:100), followed by Cy3-anti-hamster IgG (1:400) plus Cy2-streptavidin (1:400); (iv) AF488-CD4 Ab (1:50), biotin-CD8 (1:100), and Ki67 Ab (rabbit, 1:100), plus AF647-streptavidin (1:400) and Cy3-anti-rabbit IgG (1:400); or (v) biotin-CD4 Ab (1:100) or biotin-CD8 Ab (1:100) and anti-phospho-histone H3 (Ser10) Ab (rabbit polyclonal, 1:500), followed by AF647-streptavidin (1:400) and Cy3-anti-rabbit IgG (1:400). Nuclei were stained with DAPI (Invitrogen). Slides were examined with a Nikon Eclipse E800 microscope equipped with a CCD camera (Nikon, Melville, NY, USA). Percentages of CD4 and CD8 T cells expressing Ki67 and phospho-histone H3 (Ser10) were quantified on digital images taken at ×200, on five sections per allograft, with the MetaMorph Version 7.7 software.

**Multiphoton intravital microscopy.** Multiphoton intravital microscopy was performed on transplanted kidneys. After a midline incision, the transplanted kindey was carefully mobilized into a plastic cup-shaped holder attached to a x–y–z manipulator on the imaging platform. The kidney was embedded in 2% low-melting point 37 °C agarose and covered with a coverslip[45]. An Olympus FluoView FV1000 microscope equipped with a MaiTai DeepSee femtosecond-pulsed laser (Spectra-Physics) or a A1R MP Nikon Multiphoton system with a Chameleon IR Laser, a Prior ProScan III stage and a MCL Nano-Drive objective Z-drive. Lasers were tuned and mode-locked to 920 nm (GFP, YFP) or 850 nm (CellTrackerRed plus fluorescent protein excitation). Both microscopes are equipped with 4 non-descanned photomultiplier tubes with the following filter sets: Ch1 (480/40), Ch2 (540/40), Ch3 (610/80) and Ch4 (705/90). Microscope data were acquired with Olympus Fluoview v3.1 or Nikon NIS-Elements v4.3. Mice were anaesthetized with isoflurane and oxygen and core body temperature maintained at 37 °C with a homeothermic controller (TC-1000, CWE, Ardmore, PA). Animals were kept hydrated by injecting 1 ml 5% dextrose lactated ringer's solution s.c. every 60 min. Blood vessels were visualized by injecting Evans Blue (3–6 µl of 5 mg ml$^{-1}$ stock solution (15–30 µg) diluted in PBS i.v.). The kidney graft was extraverted from the abdominal cavity with intact vascular connection and immobilized in a custom cup mount. A coverslip was placed on top of the kidney and z stacks were visualized with a ×25 water immersion objective (numerical aperture: 1.05) 25 µm to 50 µm below the kidney capsule. A total of 11 slices per stack were acquired at a step size of 2.5 µm. Brightness and laser power were adjusted based on the imaging depth and kept below phototoxic levels. Dwell time was set to 8 µs per pixel (Olympus) or ×16 averaging (Nikon resonance scanner) at a resolution of 512 × 512 pixel. Approximately 30-s-long stacks were repeatedly scanned for a maximum imaging time of ~30 min per location. Up to five different locations per kidney graft were imaged. All acquired movies were analysed using Imaris software (Bitplane). Drift was corrected using the blood vessels as a reference point. Background subtraction was performed on all channels equally. All cells expressing GFP, YFP or both were enumerated manually. In the CX$_3$CR1-GFP x CD11c-YFP mouse, the MATLAB channel arithmetics plugin was used to compensate for spillover of GFP (ch1) into the YFP channel (ch2) and create GFP only, YFP only and GFP/YFP co-expression channels. Three-dimensional (3D) surfaces of OT-I T cells were generated and the number of tracks present for >2 min was used to enumerate the number of T cells. Cells were determined to be extravascular if the majority or all of the cell body had moved outside the capillary lumen.

**Ex vivo assay of antigen presentation.** Leukocytes from four B6 (CD45.1) mice transplanted 7 days before with BALB/c (CD45.2) hearts were pooled and labelled with fluorochrome-conjugated CD3, CD19, NK1.1, CD11b, CD11c, CD45.1, CD45.2, Ly6C and Ly6G Abs (all at 1:100). The Lin$^{neg}$ (that is, CD3, CD19, NK1.1) Ly6G$^{neg}$ CD45.2$^{neg}$ CD45.1$^+$ recipient-derived leukocytes were sorted on a FACSAria flow cytometer (BD-Biosciences) into CD11c$^+$ Ly6C$^{lo}$, CD11c$^{neg}$ Ly6C$^{lo}$ and CD11c$^{neg}$ Ly6C$^{hi}$ subsets (all CD11b$^+$). Sorted cells were then γ-irradiated and used as stimulators of CFSE (7.5 µM)-labelled BALB/c, 1H3.1, 2C or B6 (control) splenic naive T cells (T-cell enrichment columns, R&D Systems) in 96-well round-bottom plates (25,000 APCs: 200,000 T cells per well). After 5 days, T cells were labelled with fluorochrome-conjugated CD4 and CD8 Abs (both at 1:100), and CFSE-dilution determined by flow analysis.

**ELISA.** Detection of IFN-γ, TNF-α and IL-12p70 in cell culture supernatants was done following manufacturer's instructions (BD-PharMingen, R&D).

**Adoptive transfer of monocytes.** Ly6C$^{hi}$ and Ly6C$^{lo}$ monocytes (both CD11b$^+$ CD11c$^{neg}$) were FACS-sorted from bone marrow of B6 (CD45.1) mice. The Ly6C$^{hi}$ cells were left unlabelled and the Ly6C$^{lo}$ monocytes were CFSE (7.5 μM)-labelled. Both monocyte subpopulations were co-transferred i.v. at 1:1 ratio to B6 (CD45.2) hosts (250,000 of each subset per mouse) that were transplanted 5 days earlier with BALB/c (CD45.2) hearts. Two days later (day 7) recipients were killed and heart allografts and spleens were disaggregated, cells labelled with fluorochrome-conjugated Abs specific for CD11b, CD11c, CD45.1, CD45.2, IA$^b$ and Ly6C (all at 1:100), and analysed by flow cytometry.

**Assessment of T-cell engraftment.** Effector/memory (CD44$^{hi}$) T cells (B6, CD45.1) primed against BALB/c antigens were high-speed sorted, transferred i.v. to splenectomized CD11c-DTR-B6 bone marrow→LTβR$^{KO}$-B6 chimeras (CD45.2), and assessed by flow cytometry in peripheral blood and bone marrow 21 days later in mice injected i.p., or not, with DT (4 ng g$^{-1}$ body weight) every 2 days, starting 1 day before T-cell injection. On day 21, mice were killed and the number of bone marrow cells and peripheral blood leukocytes were counted using a hemocytometer or an automated cell counter, respectively. RBCs were lysed with NH$_4$Cl solution. Blood and bone marrow cells were labelled with CD3 (1:100), CD44 (IM7, 1:250), CD45 (30-F11, 1:100) and CD45.1 Abs and analysed by flow cytometry. For detection of eGFP$^+$ DCs, bone marrow cells were labelled with CD11c and MHC-II (IA$^b$) Abs, and analysed by flow cytometry. For *ex vivo* detection of intracellular IFN-γ in the adoptively transferred effector/memory (CD44$^{hi}$) T cells, bone marrow cells were incubated with donor (BALB/c) or syngeneic (B6) splenocytes (1:1 cell ratio, 96-well plates) in complete medium with 1 μl ml$^{-1}$ of Golgiplug (BD). After 16 h, cells were labelled with CD3, CD44 and CD45.1 Abs, fixed with 4% paraformaldehyde, permeabilized with 0.1% saponin in PBS, labelled with IFN-γ Ab and analysed by flow cytometry.

**Statistical analysis.** GraphPad Prism was used for statistical analyses. Results are expressed as mean ± s.d. of biological replicates unless otherwise specified. Comparison between two groups was performed by two-sided Student's *t*-test for unpaired samples. Multiple comparisons on a single data set were done by one-way analysis of variance, followed by Tukey–Kramer multiple comparison test. Graft survival was compared by Kaplan–Meier analysis and the log-rank test. *P* value <0.05 was considered significant. Sample sizes (biological replicates = mice in each group) were determined based on extensive published data in mouse transplantation showing statistical significance in similar group sizes with normal variation and similar variance between groups. Heart grafts not beating 48 h after transplantation were deemed technical failures and were excluded from analysis. This occurred in ~5% of cases. Kidney grafts showing poor blood perfusion at time of imaging were excluded. This occurred in ~20% of cases. No randomization was used as all mice were used were genetically defined, inbred mice. Flow and image analysis were blinded and performed by multiple independent investigators (MHO, ADH and RH). Histopathological analysis was blinded and performed by AEM.

**Data availability.** The authors declare that the data supporting the findings of this study are available within the article and its Supplementary Information Files, or from the corresponding authors on a reasonable request.

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

## Acknowledgements

This work was supported by grants from the US National Institutes of Health (R01 AI049466 and R01 AI099465 to F.G.L., and R01 HL130191 to A.E.M.); American Society of Nephrology (Merrill Grant in Transplantation to MHO and Donald E. Wesson Research Fellowship to KMY); American Heart Association (14GRNT19810000 to A.E.M.); the National Natural Science Foundation of China NSFC81401318 (to Q.L.) and the Frank & Athena Sarris Chair in Transplantation Biology to F.G.L.

## Author contributions

Q.Z., Q.L., S.J.D., Q.Z., K.M.Y., A.D.H., D.M.R.-C, A.N., W.J.S., A.L.W., R.H., R.A.H. and M.H.O. conducted the experiments and data analysis, W.D.S. designed the experiments and edited manuscript, M.H.O., F.G.L. and A.E.M. designed experiments, analysed the data and wrote the manuscript.

## Additional information

**Competing financial interests:** The authors declare no competing financial interests.

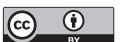

