## [Peer Review File · Nature Communications]

Reviewers' Comments:

Reviewer #1 (Remarks to the Author)

The manuscript by Zhuang and colleagues characterizes in depth the phenotype, origin and role of mononuclear phagocytes infiltrating murine cardiac and renal allografts. The authors demonstrate that recipient dendritic cells derived from non-classical monocytes rapidly replace donor dendritic cells, that they form cognate interactions with T cells, and that their presence in the allograft is required for rejection by antigen-experienced T cells even in the absence of secondary lymphoid organs. This implies that activated alloreactive T cells need to interact with host DCs in the graft, that are presenting either processed alloantigens indirectly or intact donor MHC via semi-direct presentation. This concept of a second activation checkpoint in the graft that sustains alloimmunity had been inferred from old experiments showing that indirect alloreactive CD4⁺ T cells could reject allografts even though they could not recognize graft cells directly (suggesting that they were recognizing host APCs in the graft), but it had not previously been visualized or proven. As such, this manuscript is quite important for the field. Moreover, the experiments are elegant, well-controlled, convincing and state of the art, combining mixed bone marrow chimeras, 2 photon microscopy and conditional, temporal, deletion of host DCs. The concerns are very minor and just suggestions for discussion rather than requests for additional experiments.

- In Figure 1a, the significance is stated for the increase in numbers of host DCs in the graft, but not for the decrease of numbers of donor DCs. Please add the statistical evaluation to the middle panel.

- When graft-infiltrating DCs are used to stimulate alloreactive T cells in Figure 1c, it is not clear if the gating excluded donor DCs. It is true that at day 7 post-transplantation the majority of the graft DCs are of host origin, but there may be enough donor DCs left to stimulate alloreactive T cells. Please clarify the gating for sorting and discuss supporting evidence that the authors may have further supporting that the presentation is by host and not residual donor DCs.

- The authors suggest that T cell alloreactivity increases host DC accumulation in the graft because DC accumulation is reduced in animals treated with CsA or in mice transplanted with syngeneic grafts. Do they speculate that T cell priming in secondary lymphoid organs activates host DCs that can then migrate to the graft? If so, in the splenectomized LTbR-KO model in which activated T cells are adoptively transferred, how are the host DCs attracted to the graft since DC activation in secondary lymphoid organs cannot take place in this model? Is there a T cell-independent recruitment of host DCs to the graft or can intra-graft activated T cells also recruit DCs? In other words, does CsA treatment in that model also reduce host DC recruitment to the graft?

- In Figure 4, the authors plot contact time in Figure 4e, but mean velocity and arrest coefficient in Figure 4f. It would be nice to look at the same parameter(s) in both experiments (if DCs are class I deficient versus if animals are depleted of host DCs).

Reviewer #2 (Remarks to the Author)

NCOMMS-16-02356; Nature Communications. Zhuang and Liu et al.

This manuscript reports a study on the origin and phenotypes of dendritic cells (DC) in allografts. It concludes that eliminating recipient DC at the later stage of alloimmune response can abrogate ongoing graft rejection. The paper may be of broad interest to the field, in particular with the implication of the results to transplant tolerance. A particular strength of the study is the reductionist approach employing the immunodeficient mouse model that lacks all secondary lymphoid organs which enables the study to determine cause and effect on the role of DC and effector/memory cells in ongoing graft rejection after the priming/initiation phase of alloimmunity. However, the potential of the reductionist model is not fully realized. The key conclusion needs the support from further experimentations with more rigorous controls and a clearer layout to test the impact of DC depletion on effector/memory cell-mediated graft damage in later phase of

alloimmunity, after alloimmune damage of grafts becomes evident.

Specific comments:

1. Fig 6. The reductionist approach is aimed to "drive home" the most definitive evidence for the key conceptual claim of this paper: depletion dendritic cells can interrupt ongoing graft rejection by abrogating cognate interaction between effector/memory T cells and recipient dendritic cells in the graft. There are two major concerns on the design of this experiment.

A) The most stringent control group is missing. Conditional depletion of diphtheria toxin receptor (DTR) expressing cells requires careful dosing and control to avoid the complication of unexpected toxicity, especially since different mouse models and conditions may have different sensitivity of unknown/unintended off-target cells. Therefore, there is a need for a side-by-side control group that does not have the transgenic DTR but matches all other conditions of the experimental group, including the same regimen of DT treatment. In this case, that control group would be the mice receiving LT β KO CD11c-DTR-negative (as opposed to LT β KO CD11c-DTR-positive) bone marrow but had the same procedures including splenectomy, transplant, effector/memory T cell transfer and DT treatment. In that way readers can be unequivocally convinced that DT did not have unintended, substantial effect on the allo response and graft rejection process in this model, other than depleting DC. It is true that the experiment setting requires complex and major procedures and the costly control group will likely turn out as expected. However, it is important to have the most stringent control to support a key claim that has important implication, so readers can be assured on the robustness of the data.

B) Apparently, DC depletion was administered the same day when the effector/memory cells were transferred. If DT treatment is delayed until after a time point when there is clear evidence that graft rejection has begun, the data would be stronger to support the authors' key conclusion on the role of DC in ongoing graft rejection by effector / memory cells.

2. Fig 5a. The control group has n=3. Although it is statistically significant, readers may find n=3 a bit short from the typical accepted minimum n=4-5, especially since it is the only control group in such a key experiment. Of course, each transplant experiment requires major efforts and is expensive. In Fig 5b, an additional control group, CD11c-DTR B6 chimera without DT was shown. If graft survival data in that setting are available, those data can be addition to Fig 5a to strengthen the control without additional experiments.

3. Fig 5 and Fig 6 have a major difference in that T cells are abundantly present in the graft shown in Fig 5 but not in the graft shown in Fig 6. Are the graft-infiltrating T cells shown in Fig 5 primed in lymph nodes by cells other than DC after the DC depletion? Such a speculation may be beyond the scope of experimentations in this study but worth to be discussed. Fig 6 showed a major reduction of T cells in the graft. Were the effector/memory T cells comparable in non-graft location such as in the bone marrow and blood circulation, in the presence of allograft with DC depletion? Supp Fig 3 helps a bit but it only showed comparable homeostasis in the absence of allografts.

4. Fig 4d. "OT-1 cells / volume" can be confusing to readers. Since the cells expressing GFP and YFP were manually counted as it is described in the methods, it would be more straightforward to present % of OT-1 cells contacting H-2K+ vs H-2K- DC (similar to Fig 4c). It is interesting to see that there is still substantial interaction of OT-1 T cells with DC even in the absence of cognate MHC (that part is particularly apparent in the Supp movie 1), which might worth noting in discussion.

5. Fig 3a and b. The y axis label, "DC volume" can be confusion, even to readers familiar with imaging studies.

6. Fig 2. minor point. For the x-axis label in Fig 2c "B6 MHC class-II (IAb)", H-2Ab is better than IAb since it would be consistent with the current nomenclature in the field and also consistent with the use of H-2 in the other figures in the paper.

7. Fig 1b. There is no Ki 67 staining in the figure, which is intended to show. The use of positive control may worth to be mentioned to assure readers that the Ki67 staining procedure and reagents indeed works in those experiments.

8. Statistical analysis: Student's t test is used for comparison between two groups. Figs 1, 2, 3, 5 and sup Fig 1 have panels with multiple groups. It is not clear if multi-group comparison, such as by ANOVA, has been done.

9. For both Fig 5 and 6, the authors conclude that the depletion of DC in the graft is the cause of extended graft survival. The duration of the DT treatment is not clearly stated. Is it continuous every other day until the end point of the graft? Or is it 21 days as described in supp Fig 3, and then stop? This is important since the conditional depletion in DT/DTR system is transient and DC population should recover in a few days after DT treatment is discontinued. The authors interpreted DC depletion as the cause of extended graft survival, which is understandable from the "Occam's razor" principle in scientific reasoning. However, if grafts survive long after DT is discontinued, one may argue that newly generated DC, presumably less mature, delivered a tolerogenic effect. The actual testing of such possibilities might be beyond the scope of this study, but it should be discussed. One apparent limitation of this intravital imaging platform is that it is probably not feasible to do longitudinal imaging of the same graft to gain insight on what kind of DC would repopulate the same graft after DC depletion.

10. After all, if the authors believe that DC depletion can indeed interrupt ongoing graft rejection and save the grafts or substantially extend graft survival, such a message has a major implication in clinical translation, regardless how exactly it works. Perhaps, the challenge (to both the authors and reviewers) posed by the new initiative of transparent peer review can be taken as an assurance to the authors for its likely outcome of improved robustness, fairness and open-mindedness, such that a novel observation can be addressed in a context of caveats and limitations known or known unknown for a complex in vivo process like allograft rejections, without worry of dissuading peer reviewers and editors for its conceptual and/or translational impact. As a result, readers can get a better "picture" on what is discovered and what it would take to use the discovery to drive the field forward.

PS to the authors and editors: As an author and a reviewer who has gone through the traditional peer review system with a lot of mixed feeling, I am enthused by the initiation of a transparent peer review system. It is my sincere hope that the principle of transparency will make a difference on fairness for all of us in the scientific community. With that, I am daring myself to sign my name for the first time in a peer review report: Zhibin Chen, University of Miami.

Title: Graft-infiltrating host dendritic cells play a key role in organ transplant rejection.

Ms. #: NCOMMS-16-02356

Point-by-point answers to the reviewers' comments.

Reviewer #1:

Query #1: In Figure 1a, the significance is stated for the increase in numbers of host DCs in the graft, but not for the decrease of numbers of donor DCs. Please add the statistical evaluation to the middle panel.

Answer: Two-way ANOVA analysis of the donor DC data showed significant change in DC number over time in all the groups; however, no significant differences were detected between the groups. This is now indicated in **revised Figure 1a** (* or ns) and in the **revised figure legend**.

Query #2: When graft-infiltrating DCs are used to stimulate alloreactive T cells in Figure 1c, it is not clear if the gating excluded donor DCs. It is true that at day 7 post-transplantation the majority of the graft DCs are of host origin, but there may be enough donor DCs left to stimulate alloreactive T cells. Please clarify the gating for sorting and discuss supporting evidence that the authors may have further supporting that the presentation is by host and not residual donor DCs.

Answer: We clarified this point in the first section of the Results (on page 6) and the on line Material & Methods (on page 22). The FACS-sorted DCs used to stimulate the alloreactive T cells (Fig. 1c) were within the CD45.1⁺ CD45.2^{neg} single cell gate, which includes cells of recipient origin (i.e. CD45.1⁺ cells) and excludes donor - recipient cell doublets (i.e. CD45.1⁺ CD45.2⁺ cells). The CD45.2 (donor) Ab was originally included within the label Lineage (Lin) Abs of the vertical axis of the left dot plot in the original Fig. 1c. To address the reviewer's comment, such axis has been labeled as Lin, Ly6^G & CD45.2 in the **revised Fig.1 c**.

Of note, the FACS-sorted recipient DCs were also CD11c^{int} Ly6C^{low}, a surface phenotype that differentiates these monocyte-derived DCS from donor-derived conventional DCs from the allografts that are CD11c^{high} Ly6C^{neg}.

As mentioned in the Results, our findings demonstrate that donor DCs represent less than 5% of the DCs in cardiac allografts by day 7 after surgery. Thus, even without excluding by FACS-sorting the donor DCs from the recipient DCs on day 7, the donor DCs would be at extremely low APC : T cell ratios in the MLCs depicted in Fig. 1c & e (beyond a 1 APC : 160 T cell ratio).

Query #3: The authors suggest that T cell alloreactivity increases host DC accumulation in the graft because DC accumulation is reduced in animals treated with CsA or in mice transplanted with syngeneic grafts. Do they speculate that T cell priming in secondary lymphoid organs activates host DCs that can then migrate to the graft? If so, in the splenectomized LTbR-KO model in which activated T cells are adoptively transferred,

how are the host DCs attracted to the graft since DC activation in secondary lymphoid organs cannot take place in this model? Is there a T cell-independent recruitment of host DCs to the graft or can intragraft activated T cells also recruit DCs? In other words, does CsA treatment in that model also reduce host DC recruitment to the graft?

Answer: We are not speculating that T cell priming in secondary lymphoid organs activates host DCs that then migrate to the graft. Larsen et al have in fact shown that DCs do not enter the graft from the circulation (*Transplantation* 50:294-301, 1990). Instead, based on what is shown in the current manuscript and in our previous publication (Oberbarnscheidt et al, *JCI*, 2014), we are implying that DCs in the graft arise from recruited monocytes. This monocyte recruitment can occur independent of T cells (Oberbarnscheidt et al, *JCI*, 2014) but is enhanced when effector T cells attack the graft. Therefore, in the splenectomized LTbR-KO model, the transferred effector T cells would be expected to increase monocyte migration to the graft, and therefore intragraft DC numbers, independent of secondary lymphoid organs.

Query #4: In Figure 4, the authors plot contact time in Figure 4e, but mean velocity and arrest coefficient in Figure 4f. It would be nice to look at the same parameter(s) in both experiments (if DCs are class I deficient versus if animals are depleted of host DCs).

Answer: The experiment in Figure 4e used CD11c-YFP recipients in which both DC and transferred T cells could be visualized. The experiment in Figure 4f, where recipient DC were depleted, we used CD11c-DTR.GFP into wild-type bone marrow chimeras. In these mice, the GFP reporter is very dim and cannot be visualized by 2P imaging. Therefore, T cell-DC interactions (contact time) could not be determined in Figure 4f. However, we did look at mean velocity and arrest coefficient in the experiment in Fig. 4e and the plots have been added to **revised Figure 4e**. As shown, T cells that contacted class I-deficient DCs had higher mean velocity and lower arrest coefficient than those that contacted class I-sufficient DCs. These data are congruent with the data shown in Fig. 4f.

Reviewer #2:

Query #1: Fig 6. The reductionist approach is aimed to "drive home" the most definitive evidence for the key conceptual claim of this paper: depletion dendritic cells can interrupt ongoing graft rejection by abrogating cognate interaction between effector/memory T cells and recipient dendritic cells in the graft. There are two major concerns on the design of this experiment:

A) The most stringent control group is missing. Conditional depletion of diphtheria toxin receptor (DTR) expressing cells requires careful dosing and control to avoid the complication of unexpected toxicity, especially since different mouse models and conditions may have different sensitivity of unknown/unintended off-target cells. Therefore, there is a need for a side-by-side control group that does not have the transgenic DTR but matches all other conditions of the experimental group, including the same regimen of DT treatment. In this case, that control group would be the mice receiving LT β KO CD11c-DTR-negative (as opposed to LT β KO CD11c-DTR-positive) bone marrow but had the same procedures including splenectomy, transplant, effector/memory T cell transfer and DT treatment. In that way readers can be unequivocally convinced that DT did not have unintended, substantial effect on the allo response and graft rejection process in this model, other than depleting DC. It is true that the experiment setting requires complex and major procedures and the costly control group will likely turn out as expected. However, it is important to have the most stringent control to support a key claim that has important implication, so readers can be assured on the robustness of the data.

B) Apparently, DC depletion was administered the same day when the effector/memory cells were transferred. If DT treatment is delayed until after a time point when there is clear evidence that graft rejection has begun, the data would be stronger to support the authors' key conclusion on the role of DC in ongoing graft rejection by effector / memory cells.

Answer: The additional groups proposed by the reviewer have been performed and added to **revised Fig. 6a**. The figure legend and Results section (**on page 11, second paragraph**) have been revised accordingly. We found that DT administration to splenectomized WT to LT β RKO chimeras does not delay allograft rejection mediated by transferred effector T cells. We also found that delaying DT administration until day 10 after transplantation in splenectomized CD11c-DTR to LT β RKO chimeras delays allograft rejection by ~14 days, although this did not reach statistical significance. The latter result implies that once allograft damage has sufficiently advanced (8 days after effector T cell transfer), DC depletion becomes less effective at preventing the eventual demise of the graft. This finding does not alter our main conclusion that host-derived graft DCs play a key role in allograft rejection.

Query #2: Fig 5a. The control group has n=3. Although it is statistically significant, readers may find n=3 a bit short from the typical accepted minimum n=4-5, especially since it is the only control group in such a key

experiment. Of course, each transplant experiment requires major efforts and is expensive. In Fig 5b, an additional control group, CD11C-DTR B6 chimera without DT was shown. If graft survival data in that setting are available, those data can be addition to Fig 5a to strengthen the control without additional experiments.

Answer: Per the reviewer's request, we increased the number of recipients from 3 to 5 in the BALB/c heart Tx → WT B6 chimeras + DT control group (**revised Fig. 5a**). We also performed the second control group suggested by the reviewer, which is BALB/c heart Tx → CD11c-DTR-B6 chimera + PBS and included the data in **revised Fig. 5a**. The text of the **revised Results section on page 10, second paragraph**, notes the additional control group.

Query #3: Fig 5 and Fig 6 have a major difference in that T cells are abundantly present in the graft shown in Fig 5 but not in the graft shown in Fig 6. Are the graft-infiltrating T cells shown in Fig 5 primed in lymph nodes by cells other than DC after the DC depletion? Such a speculation may be beyond the scope of experimentations in this study but worth to be discussed. Fig 6 showed a major reduction of T cells in the graft. Were the effector/memory T cells comparable in non-graft location such as in the bone marrow and blood circulation, in the presence of allograft with DC depletion? Supp Fig 3 helps a bit but it only showed comparable homeostasis in the absence of allografts.

Answer: The presence of more T cells in Fig. 5 than Fig. 6 is most likely due to differences in the experimental design. In Fig. 5, rejection was allowed to proceed for several days (5 days) before DC depletion was initiated. In Fig. 6, DC depletion was initiated at the time of T cell transfer. In the experiment shown in Fig. 6, we did not quantify effector/memory T cells in non-graft locations. Instead, we tested the effect of DC depletion on the number and function of transferred T cells in the same model without the confounding effect of the allograft on T cell proliferation and expansion. As shown in Suppl. Fig. 3, DC depletion did not affect T cell number or function in the blood or bone marrow.

Query #4: Fig 4d. "OT-1 cells / volume" can be confusing to readers. Since the cells expressing GFP and YFP were manually counted as it is described in the methods , it would be more straightforward to present % of OT-1 cells contacting H-2K+ vs H-2K- DC (similar to Fig 4c). It is interesting to see that there is still substantial interaction of OT-1 T cells with DC even in the absence of cognate MHC (that part is particularly apparent in the Supp movie 1), which might worth noting in discussion.

Answer: **Fig. 4d** has been revised to present percent of OT-I T cells as suggested by the reviewer. We agree with the reviewer that there is still substantial interaction of OT-I cells with H-2Kb^{-/-} DCs. This could possible be due to either chemokine-mediated interactions, especially the shorter contact ones, or to cross-dressing of H-2Kb^{-/-} DCs with the H-2Kb molecule carrying the OVA peptide. As suggested by the reviewer, these possibilities have been noted in the **revised Discussion (on page13, paragraph 2)**.

Query #5: Fig 3a and b. The y axis label, "DC volume" can be confusion, even to readers familiar with imaging studies.

Answer: As evident from the day 7 image, the DC infiltrate was very dense so that individual DC could not be identified. Quantitation of graft-infiltrating DC by cell number in Fig. 3a was therefore not possible. Instead, we calculated the voxel volume represented by the fluorescent signal of *total* recipient infiltrating DC (volume), allowing comparison of DC infiltration at different time points. To avoid confusion, we have changed the y-axis label in **revised Fig. 3a** to "Total DC volume" instead of "DC volume". In Fig. 3b, where the data were generated by flow analysis, we show actual "DC number" on the y-axis rather than total "DC volume".

Query #6: Fig 2. minor point. For the x-axis label in Fig 2c "B6 MHC class-II (IAb)", H-2Ab is better than IAb since it would be consistent with the current nomenclature in the field and also consistent with the use of H-2 in the other figures in the paper.

Answer: We have modified the nomenclature used in the histogram of **revised Fig 2C**, as requested.

Query #7: Fig 1b. There is no Ki 67 staining in the figure, which is intended to show. The use of positive control may worth to be mentioned to assure readers that the Ki67 staining procedure and reagents indeed works in those experiments.

Answer: The green circular areas in Fig 1b are Ki67⁺ nuclei of CD11c^{neg} cells (likely T cells) next to CD11c⁺ DCs (in red), the former cells serve as the endogenous positive control for Ki67 labeling requested by the reviewer. We clarified this issue in the **revised Fig. 1b legend** and added two insets in **revised Fig 1b** showing in detail the Ki67⁺ cells as endogenous controls for Ki67 labeling.

Query #8: Statistical analysis: Student's t test is used for comparison between two groups. Figs 1, 2, 3, 5 and sup Fig 1 have panels with multiple groups. It is not clear if multi-group comparison, such as by ANOVA, has been done.

Answer: We have corrected the analysis of single data sets for multiple comparisons in Figures 1, 2 and 5, and in Supplementary Figure 1 by using 1-way ANOVA. The corrected analysis showed significant differences that are consistent with the original conclusions. In Figure 3, we left the analysis by Student's t test because all comparisons were done between 2 groups. The **Methods Section on page 24, last paragraph**, has been revised to note the change in the statistical methods used.

Query #9: For both Fig 5 and 6, the authors conclude that the depletion of DC in the graft is the cause of extended graft survival. The duration of the DT treatment is not clearly stated. Is it continuous every other day until the end point of the graft? Or is it 21 days as described in sup Fig 3, and then stop? This is important

since the conditional depletion in DT/DTR system is transient and DC population should recover in a few days after DT treatment is discontinued. The authors interpreted DC depletion as the cause of extended graft survival, which is understandable from the "Occam's razor" principle in scientific reasoning. However, if grafts survive long after DT is discontinued, one may argue that newly generated DC, presumably less mature, delivered a tolerogenic effect. The actual testing of such possibilities might be beyond the scope of this study, but it should be discussed. One apparent limitation of this intravital imaging platform is that it is probably not feasible to do longitudinal imaging of the same graft to gain insight on what kind of DC would repopulate the same graft after DC depletion.

Answer: In both cases (Fig 5 and 6), DT treatment was continuous every other day until the endpoint of the experiment (graft rejection). We added this information to the **revised Results section, on page 10, paragraph 2** and **on page 11, paragraph 2**. This regimen causes persistent depletion of DCs.

Query #10: After all, if the authors believe that DC depletion can indeed interrupt ongoing graft rejection and save the grafts or substantially extend graft survival, such a message has a major implication in clinical translation, regardless how exactly it works. Perhaps, the challenge (to both the authors and reviewers) posed by the new initiative of transparent peer review can be taken as an assurance to the authors for its likely outcome of improved robustness, fairness and open-mindedness, such that a novel observation can be addressed in a context of caveats and limitations known or known unknown for a complex in vivo process like allograft rejections, without worry of dissuading peer reviewers and editors for its conceptual and/or translational impact. As a result, readers can get a better "picture" on what is discovered and what it would take to use the discovery to drive the field forward.

Answer: We thank the reviewer for the insightful comments.

Reviewers' Comments:

Reviewer #1 (Remarks to the Author)

Please provide the source and dose of Cyclosporin A used in Figure 1.

Reviewer #2 (Remarks to the Author)

In this revised manuscript (NCOMMS-16-02356), the authors have addressed all concerns raised in the previous critiques.

Title: Graft-infiltrating host dendritic cells play a key role in organ transplant rejection.

Ms. #: NCOMMS-16-02356A

Point-by-point answers to the reviewers' comments.

Reviewer #1:

Query #1: Please provide the source and dose of Cyclosporin A used in Figure 1.

Answer: The information requested has been added to the Methods section (under the subheading Reagents).

Reviewer #2:

In this revised manuscript (NCOMMS-16-02356), the authors have addressed all concerns raised in the previous critiques.